# backbone: An R package to extract network backbones

**Zachary P. Neal** (ID) *

Psychology Department, Michigan State University, East Lansing, MI, United States of America

* zpneal@msu.edu

**Data Availability Statement:** All files needed to replicate these examples are available at https://osf.io/8tuc7/.

**Funding:** ZN received funding from the National Science Foundation (www.nsf.gov), awards #1851625 and #2016320. The funders had no role

## Abstract

Networks are useful for representing phenomena in a broad range of domains. Although their ability to represent complexity can be a virtue, it is sometimes useful to focus on a simplified network that contains only the most important edges: the backbone. This paper introduces and demonstrates a substantially expanded version of the backbone package for R, which now provides methods for extracting backbones from weighted networks, weighted bipartite projections, and unweighted networks. For each type of network, fully replicable code is presented first for small toy examples, then for complete empirical examples using transportation, political, and social networks. The paper also demonstrates the implications of several issues of statistical inference that arise in backbone extraction. It concludes by briefly reviewing existing applications of backbone extraction using the backbone package, and future directions for research on network backbone extraction.

## Introduction

Networks are useful for representing phenomena in a broad range of domains [1, 2]. Although their ability to represent complexity can be a virtue, in some cases (e.g., computationally intensive analysis, presence of noise, visualization) it is useful to simplify a network and focus instead on its *backbone*. Given a complex network $\mathbf{N}$, which may be weighted or unweighted, its backbone $\mathbf{N}'$ is a sparse and unweighted subgraph that aims to preserve or reveal important structural features. Many backbone methods have been proposed for extracting a backbone $\mathbf{N}'$ from a network $\mathbf{N}$, with different methods designed for different types of networks, or to preserve different types of structural features. However, applying these methods in practice has been challenging because they were either not implemented or implemented in different software languages.

The `backbone` package for R aims to overcome this practical limitation of network backbone extraction by providing an integrated implementation of existing methods. The formal mathematical details of these methods, and evidence of their performance as structure-preserving or structure-revealing backbone models, is extensively documented elsewhere and referenced below. Instead, the purpose of this paper is to provide a practical guide to using the `backbone` package for backbone extraction. It is organized in six sections. The first section provides an overview of the `backbone` package. The second, third, and fourth sections

in study design, data collection and analysis, decision to publish, or preparation of the manuscript.

**Competing interests:** The authors have declared that no competing interests exist.

illustrate how the package can be used to extract the backbone from a weighted network, from a bipartite projection, and from an unweighted network, respectively. Each of these sections begin with an overview of the relevant backbone models, then provide a small toy example to illustrate the model, followed by an empirical example to demonstrate its use in practice. The fifth section reviews issues of statistical inference that arise in backbone models. Finally, the sixth section concludes with a discussion of past applications of the backbone package, its limitations, and future directions for the implementation of backbone models.

## The backbone package

The most recent release of the backbone package can be installed in R [3] from the Comprehensive R Archive Network (CRAN) using:

```
> install.packages("backbone")
trying URL 'https://cran.rstudio.com...'
Content type 'application/x-gzip' length 1758582 bytes (1.7
MB)
==================================================
downloaded 1.7 MB
```

Once installed, the backbone package can be loaded using:

```
> library(backbone)
_____     backbone v2.1.0
| _ \   Cite: Neal, Z. P., (2022). backbone: An R package to
extract network
|#|_) |    backbones. PLOS ONE.
|# _ <
|#|_) |Help: type vignette("backbone"); email zpneal@msu.edu;
github zpneal/backbone
|_____/ Beta: type devtools::install_github("zpneal/back-
bone", ref = "devel")
```

The startup message displays the version of the backbone package that is installed and has been loaded for use. It also displays the recommended citation for the package, sources for help using the package, and the command to install the beta release of the backbone package. Additional information about the CRAN distribution is available at https://CRAN.R-project.org/package=backbone, while additional materials relating to backbone, including papers, presentations, workshop materials, and data sets are available at http://www.zacharyneal.com/backbone. The code and data necessary to replicate the examples shown in this paper are available at https://osf.io/8tuc7/.

Fig 1 illustrates the typical workflow of the backbone package. A user begins with source data, which can take the form of an R matrix object, sparse Matrix object, an edgelist stored as a dataframe object, or an igraph object. The source data may represent an unweighted bipartite network, a weighted unipartite network, or an unweighted unipartite network. The relevant backbone models depend on the type of network. For example, when starting with an unweighted bipartite network, a user may extract the backbone from its weighted projection using any of the following models: sdsm(), fdsm(), fixedrow(), fixedcol(), fixedfill(), disparity(), or global(). Extracting the backbone using one of these functions yields an unweighted unipartite network, thus as the figure illustrates, the user may subsequently extract a second-order backbone using the sparsify() function.

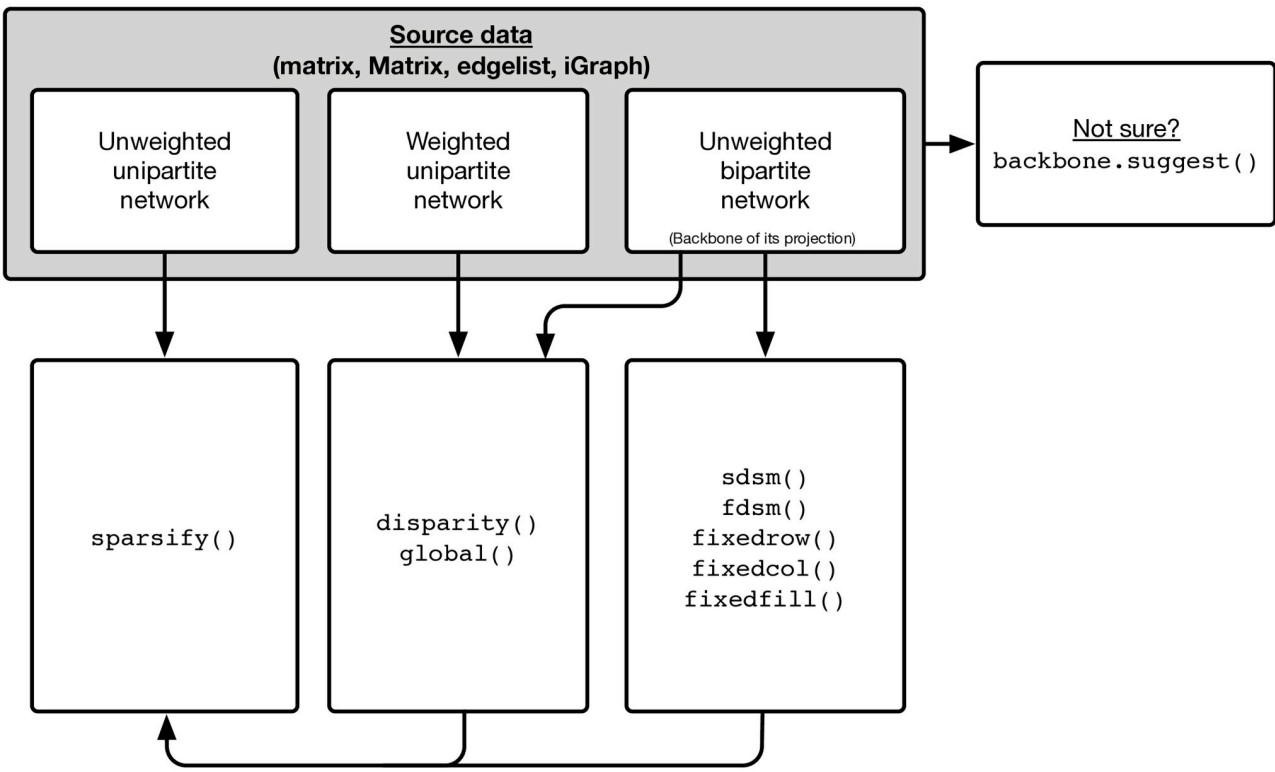

**Fig 1. Workflow of the backbone package.**

The `backbone.suggest()` function can be used to examine the source data, identify the relevant backbone models, and suggest the most appropriate model. For example:

```
> N <- matrix(runif(100), 10, 10)
> backbone.suggest(N)
The disparity filter is suggested. Type "?disparity" for more
information.
```

Here, the source data is a randomly generated $10 \times 10$ matrix of values between 0 and 1. Based on these characteristics, it is recognized as a weighted and directed adjacency matrix representing a weighted unipartite network. Therefore, the disparity filter model is recommended, for which additional information is available by typing `?disparity` in the R console.

In the sections below, we use randomly generated data like this for toy examples, but also use real data for empirical examples. The empirical examples use data on air transportation in the US [4], bill sponsorship in the US Senate [5], and friendships among faculty at a UK university [6]. A random number seed can be set to ensure that the randomly generated data are replicable, and the empirical data can be loaded, using:

```
> set.seed(1)
> load("backbone2_tutorial.Rdata")
```

## Backbones of weighted networks

### Background

In a weighted network, each edge has a weight that captures the strength of the relationship between the two nodes it connects. These weights can represent capacities when larger values

capture stronger relations, or can represent costs when larger values capture weaker relations. The precise meaning of these weights depends on the context of the network, but the models implemented in the `backbone` package assume they represent capacities (i.e., stronger relations have larger weights). In a social network, the weights might represent the strength of a friendship, or the frequency of communication between two people. Alternatively, in a transportation network the weights might represent the number of people flying from one city to another, or the capacity of a road to carry vehicles. These weights provide some information about the importance of edges in the network, and backbone models designed for weighted networks use this information to determine which edges should be preserved in the backbone.

The simplest backbone model applicable to weighted networks is the *global threshold*. The global threshold model preserves all edges whose weight exceeds a specified threshold. It can be classified as a 'global' model because it applies the same criteria to all edges in the network, and can be classified as a 'structural' model because it uses only structural information (here, the edges' weights). While it can be appealing for its simplicity, the global threshold model can be problematic when applied to networks in which the edge weights follow a long-tailed distribution (e.g., mostly weak edges, a few strong edges) and networks in which different parts of the network have different characteristic edge weights (e.g., multi-scale networks).

Many alternative backbone models have been developed for weighted networks exhibiting these characteristics [7–21], however among the most widely used is the *disparity filter* [4]. The disparity filter compares an edge's observed weight to its expected weight in a null model where a node's total weight is uniformly distributed across its edges. For example, it might compare the observed number of people traveling between New York and Los Angeles (a large number) to the number that would be expected if each of New York's passengers randomly selected a destination (likely a much smaller number). An edge is preserved in the backbone if its observed weight is statistically significantly (at a specified $\alpha$ significance level) stronger than expected under such a null model. This model can be classified as a 'local' model because an edge's importance is evaluated using information in its neighborhood, and can be classified as a 'statistical' model because it makes inferences about edge importance by reference to a statistical null model.

## Toy example

To illustrate extracting the global threshold and disparity filter backbones from weighted networks using the `backbone` package, we begin with a simple toy example:

```
> mat <- matrix(c(0,10,10,10,10,75,0,0,0,0,
                  10,0,1,1,1,0,0,0,0,0,
                  10,1,0,1,1,0,0,0,0,0,
                  10,1,1,0,1,0,0,0,0,0,
                  10,1,1,1,0,0,0,0,0,0,
                  75,0,0,0,0,0,100,100,100,100,
                  0,0,0,0,0,100,0,10,10,10,
                  0,0,0,0,0,100,10,0,10,10,
                  0,0,0,0,0,100,10,10,0,10,
                  0,0,0,0,0,100,10,10,10,0),10)
```

This network is shown in the left panel of Fig 2, where stronger edges are drawn using thicker lines. The edge weights in this network clearly follow a long-tailed distribution: a few edges are very strong (weights ≥75), while the majority of edges are fairly weak (weights ≤10). Additionally, the network has a clear multi-scale structure: the cluster in the lower left is

**Weighted Original**   **Global Threshold**   **Disparity Backbone**

**Fig 2. Extracting the backbone of a weighted network (toy example).**

characterized by relatively strong edges, while the cluster in the upper right is characterized by relatively weak edges.

We can extract a global threshold backbone from this network using the `global()` function:

```
> unweighted <- global(mat, upper = function(x)mean(x), class = "igraph")
```

In this example, the `global()` function takes three arguments: the source network, the upper threshold for edges to discard, and the desired class of the result. Here, the source network takes the form of an adjacency matrix `mat`. The upper threshold for edges to discard is defined as the mean of all edge weights, thereby yielding a backbone of stronger-than-average edges. Finally, the result is returned as an `igraph` object, to facilitate subsequent visualization and analysis. The global threshold backbone is shown in the middle panel of Fig 2, which highlights its shortcomings when applied to a multiscale network. By applying the same threshold to all edges, it preserves edges in the high-weight cluster, but ignores the edges in the low-weight cluster.

We can extract a disparity filter backbone from this network using the `disparity()` function:

```
> backbone <- disparity(mat, alpha = 0.05, class = "igraph")
```

The `disparity()` function also takes the adjacency matrix `mat` as input and returns the backbone as an `igraph` object, however it does not use a global threshold value. Instead, the user specifies `alpha`, the statistical significance level. In this example, an edge is preserved if its weight is larger than the weight of the corresponding edge it at least 95% of null model networks. The disparity backbone is shown in the right panel of Fig 2, which illustrates its ability to preserve the original network's hub-and-spoke structure despite its multi-scale edge weights.

## Empirical example

The toy example illustrates the basic operation of the `global()` and `disparity()` backbone extraction functions. To illustrate how these backbone extraction functions might be used in practice, we apply them to extract the backbone of the US air transportation network, which is known to have a hub-and-spoke structure containing many low-degree nodes (e.g., most airports serve few passengers) and a few very high-degree nodes (e.g., hub airports that serve many passengers). These data closely resemble the airline data used in the initial demonstration of the disparity filter [4], and were obtained from the US Bureau of Transportation Statistics' Airline Origin and Destination Survey, which contains a 10% random sample of all

domestic airline tickets in 2019. The weighted, symmetric network records the number of passengers traveling between 382 airports in the continental US. We can inspect the entries in this network:

```
> airport["JFK","LAX"]   #Passengers between New York and Los
Angeles
[1] 326756
> airport["LAN","GRR"]   #Passengers between Lansing and
Grand Rapids
[1] 94
> airport["LAN","LAX"]   #Passengers between Lansing and Los
Angeles
[1] 0
```

For example, we observe that many passengers fly between JFK (New York) and LAX (Los Angeles), while very few fly between LAN (Lansing, Michigan) and GRR (Grand Rapids, Michigan) and none fly between LAN and LAX because there is no scheduled service.

The weighted original network is shown in the left panel of Fig 3, which highlights that the network's high density obscures any particular structure. The network has a very large mean weighted degree $\langle k \rangle$ = 374215 because some airports such as Atlanta's Hartsfield-Jackson serve very large volumes of passengers. When a network has a hub-and-spoke structure, the probability of observing a node with degree $k$ is approximately distributed as $k^\gamma$, where typically $2 < \gamma < 3$ [22]. However, in this network $\gamma$ = 1.15 [23], which falls outside this range and suggests the network lacks the expected hub-and-spoke structure.

We can extract a global threshold backbone from this network using:

```
> unweighted <- global(airport, upper = function(x)mean(x),
class = "igraph")
```

The resulting backbone, which preserves only stronger-than-average edges, is shown in the middle panel of Fig 2. It is clearly much sparser, with a smaller mean degree $\langle k \rangle$ = 14.81. However, it remains fairly dense, and is dominated by the high-volume routes on the East coast. Its degree scaling exponent $\gamma$ = 1.83 remains inconsistent with a hub-and-spoke transportation network [22, 23].

We can extract a disparity filter backbone from this network using:

```
> backbone <- disparity(airport, alpha = 0.001, class =
"igraph", narrative = TRUE)
=== Suggested manuscript text and citations ===
We used the backbone package for R (v2.1.0; Neal, 2022) to
extract the unweighted backbone of a weighted and undirected
unipartite network containing 382 nodes.
```

### Weighted Original
$\langle k \rangle$ = 374215.71, $P(k) \sim k^{-1.15}$

### Global Threshold
$\langle k \rangle$ = 14.81, $P(k) \sim k^{-1.83}$

### Disparity Backbone
$\langle k \rangle$ = 4.34, $P(k) \sim k^{-2.24}$

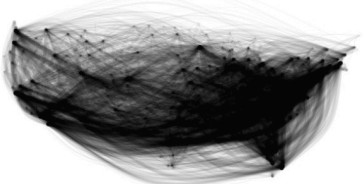 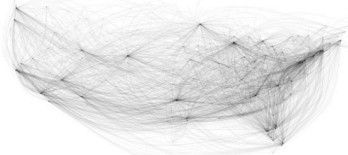 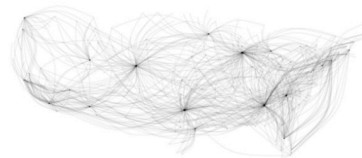

**Fig 3. Extracting the backbone of a weighted network of continental US airline traffic in 2019.**

```
   An edge was retained in the backbone if its weight was statis-
tically significant (alpha = 0.001) using the disparity filter
(Serrano et al., 2009). This reduced the number of edges by
91.4%, and reduced the number of connected nodes by 14.7%.
   Neal, Z. P. (2022). backbone: An R Package to Extract Network
Backbones. arXiv:2203.11055 [cs.SI]. https://doi.org/10.48550/
arXiv.2203.11055
   Serrano, M. A., Boguna, M., & Vespignani, A. (2009). Extract-
ing the multiscale backbone of complex weighted networks. Pro-
ceedings of the National Academy of Sciences, 106(16), 6483-
6488. https://doi.org/10.1073/pnas.0808904106
```

In this example, following [4] we use a conservative statistical significance level of $\alpha = 0.001$. We also include the narrative = TRUE argument, which automatically generates text and citations describing what the disparity() function has done, and that can be used or adapted for a manuscript. The resulting backbone, which preserves edges whose weights are unusually large compared to the null model, is shown in the right panel of Fig 3. It is quite sparse: the narrative text indicates that the number of edges was reduced by 91.4%, and we observe it has a small mean degree $\langle k \rangle = 4.34$. Its degree scaling exponent $\gamma = 2.34$ is now consistent with a transportation infrastructure known to have a hub-and-spoke organization [22, 23]. Indeed, the visualization clearly illustrates this organization, which is anchored by the hub airports of ORD (Chicago), DEN (Denver), DFW (Dallas-Fort Worth), and ATL (Atlanta).

## Backbones of bipartite projections

### Background

A bipartite projection is a special type of weighted network in which the edge weights capture the number of *artifacts* shared in common by two *agents*, and thus capture co-occurrence relations [24]. They arise in a wide range of contexts, including co-author networks where authors (the agents) share papers (the artifacts), co-sponsorship networks where legislators share bills, co-attendance networks where people share events, and co-expression networks where genes share expressed proteins. As a variety of weighted network, the backbone of a bipartite projection can be extracted using global() or disparity(). However, it is usually more appropriate to extract the backbone of a bipartite projection using a backbone model designed specifically for this purpose.

To understand why a specialized backbone model is needed for a bipartite projection, it is helpful to consider how the edge weights in a bipartite projection are defined. A bipartite network in which authors (the agents) are connected to the papers (the artifacts) they have written can be represented by an incidence matrix **B**, where where $B_{ik} = 1$ if author $i$ wrote paper $k$. We obtain the adjacency matrix of the bipartite projection **P** from the incidence matrix of the bipartite network **B** as $\mathbf{P} = \mathbf{B}\mathbf{B}'$, where each entry $P_{ij}$ indicates the number of papers co-authored by authors $i$ and $j$ (i.e., the edge weight). Applying a conventional backbone model such as the disparity filter to this bipartite projection would decide which edges to preserve based only on these edge weights, but would fail to consider two important pieces of information that are contained in the original bipartite network. First, the row sums of **B** record how many papers the $i^{th}$ author wrote (i.e., the agent degrees). This is important to consider because if two authors each wrote many papers, it is likely they would have co-authored a few just by chance. Second, the column sums of **B** record how many authors the $k^{th}$ paper has (i.e., the artifact degrees). This is important to consider because if a paper has many authors, then observing that two people are both on the author list is not particularly noteworthy.

Backbone models designed specifically for bipartite projections are unique because they are applied directly to the bipartite network, not to its projection, so that they can incorporate this information. While there are multiple such models [24–28], they all use a common approach: An edge's observed weight in the projection is compared to its expected weight in a projection obtained from a random bipartite network generated by a null model. An edge is preserved in the backbone if its observed weight is statistically significantly (at a specified $\alpha$ significance level) stronger than expected under the null model. These models differ in the constraints they impose on the random bipartite networks. In this section, we illustrate the *Stochastic Degree Sequence Model* (SDSM), which constrains the random bipartite networks to have row sums and column sums that match those of the observed bipartite network on average. The SDSM is faster than the *Fixed Degree Sequence Model*, and more accurately recovers known structural characteristics than the *Fixed Row*, *Fixed Column*, or *Fixed Fill* models [28].

## Toy example

To illustrate extracting the SDSM backbone from a bipartite projection using the `backbone` package, we begin with a simple toy example:

```
> B <- rbind(cbind(matrix(rbinom(250,1,.8),10),
        matrix(rbinom(250,1,.2),10),
        matrix(rbinom(250,1,.2),10)),
    cbind(matrix(rbinom(250,1,.2),10),
        matrix(rbinom(250,1,.8),10),
        matrix(rbinom(250,1,.2),10)),
    cbind(matrix(rbinom(250,1,.2),10),
        matrix(rbinom(250,1,.2),10),
        matrix(rbinom(250,1,.8),10)))
> sum(B[1,]) #Agent 1's degree
[1] 28
> sum(B[,1]) #Artifact 1's degree
[1] 11
```

This generates a random bipartite network, represented as an incidence matrix in `B`. This network consists of 30 agents and 75 artifacts, and is embedded with a community structure. The agents are split into three groups of 10, and the artifacts are split into three groups of 25. There is an 80% chance of an edge between an agent and artifact belonging to the same group (i.e., people are likely to attend their own group's events), and only a 20% chance of an edge between an agent and artifact belonging to different groups (i.e., people are unlikely to attend another group's events). Examining this bipartite network, we observe that Agent 1 is associated with 28 artifacts (i.e., agent degree), and that Artifact 1 is associated with 11 agents (i.e., artifact degree).

Given this bipartite network, we can construct and examine its bipartite projection:

```
> P <- B %*% t(B) #Construct bipartite projection
> P[1, 2] #Edge weight for two agents in the same group
[1] 17
> P[1, 20] #Edge weight for two agents in different groups
[1] 7
> min(P) #Smallest edge weight
[1] 3
```

The projection is constructed by multiplying the bipartite network `B` by its transpose `t(B)`. In the projection, we observe that agents 1 and 2 shared 17 artifacts in common, a large

**Weighted Original**     **Disparity Backbone**     **SDSM Backbone**

**Fig 4. Extracting the backbone of a bipartite projection (toy example).**

number because they are members of the same group. In contrast, agents 1 and 20 share only 7 artifacts in common, a small number because they are members of different groups. Notably, however, the smallest number of artifacts shared by any pair of agents is still 3. This means that all agents share at least *some* artifacts in common, which makes the bipartite projection not particularly useful as a network, and occurs because the construction of a bipartite projection "induces an inflation in the number of links" [29]. Indeed, as illustrated in the left panel of Fig 4, the weighted bipartite projection is so dense that no particular structure is visible, including the community structure that is known to exist.

Because the bipartite projection is a weighted network, we could extract its backbone using the disparity filter:

```
> disparity <- disparity(P, alpha = 0.05, class = "igraph")
This object looks like it could be a bipartite projection. If
so, consider extracting the backbone using a model designed for
bipartite projections: sdsm, fdsm, fixedfill, fixedrow, or
fixedcol.
```

The middle panel of Fig 4 shows the disparity backbone, which is empty. That is, the disparity filter is unable to identify any significant edges to preserve in the backbone. This occurs because the disparity filter is relying only on information contained in the bipartite projection, but does not use any of the information contained in the underlying bipartite network. Here, the disparity() function detects that the source network appears to be a bipartite projection, and recommends instead using a backbone model specifically designed for such a network.

Following this advice, we can instead extract a backbone using the stochastic degree sequence model and sdsm() function:

```
> backbone <- sdsm(B, alpha = 0.05, class = "igraph")
```

Like the disparity() function, the sdsm() function also takes alpha and class arguments. However, the source network is not the bipartite projection P, but the original bipartite network B from which it was constructed. The right panel of Fig 4 shows the SDSM backbone, which is sparse and clearly displays the known three-community structure.

## Empirical example

To illustrate the extraction of a bipartite projection backbone in practice, we use data on bill sponsorship patterns in the US Senate's 115[th] session (2017-2018) [30, 31]. In the US Senate, legislators can express support for a bill by 'sponsoring' it. Therefore, bipartite projections of

bill sponsorship data yield co-sponsorship networks that are often used to study political alliances among legislators. The bipartite network's incidence matrix records the bill sponsorships of 105 Senators on 3665 bills. We can inspect the details of this bipartite network:

```
> senate[1:2, 1:2] #First two rows and columns
            S.1 S.100
Sen. Alexander, Lamar [R-TN]    0     1
Sen. Baldwin, Tammy [D-WI]    0     0
> sum(senate["Sen. Stabenow, Debbie [D-MI]",]) #Sponsorships
by Sen. Stabenow
[1] 316
> sum(senate[,"S.1006"]) #Sponsors of Equality Act
[1] 48
```

Looking at the first four entries of the incidence matrix, we see that neither Senators Alexander nor Baldwin sponsored Senate Bill 1, and that Senator Alexander sponsored Senate Bill 100 while Senator Baldwin did not. Row sums indicate each Senator's number of sponsorships, for example, Senator Stabenow sponsored 316 bills in this session. Similarly, column sums indicate each bill's number of sponsors, for example, Senate Bill 1006 had 48 sponsors. This bill, known as the Equality Act, would have amended the Civil Rights Act of 1964 to prohibit discrimination by sex, sexual orientation, and gender identity, and was sponsored only by Democrats and Independents.

Given this bipartite network, we can construct and examine its bipartite projection:

```
> P <- senate %*% t(senate) #Construct bipartite projection
> P["Sen. Stabenow, Debbie [D-MI]", "Sen. Peters, Gary C.
[D-MI]"]
[1] 151
> P["Sen. Stabenow, Debbie [D-MI]", "Sen. Cruz, Ted [R-TX]"]
[1] 14
```

We see that the Senators Stabenow and Peters co-sponsored 151 bills together, reflecting their likely alliance as members of the same party and representatives of the same state. In contrast, we see that Senators Stabenow and Cruz sponsored only 14 bills together, reflecting their lack of partnership as members of opposing parties and representatives of different states. Notably, however, despite the sharp ideological differences between Stabenow and Cruz, they still co-sponsored *some* bills together and are still connected in the bipartite projection.

The weighted bipartite projection is shown in the left panel of Fig 5; Democrats and Independents are shown in blue, while Republicans are shown in red. It illustrates several problems

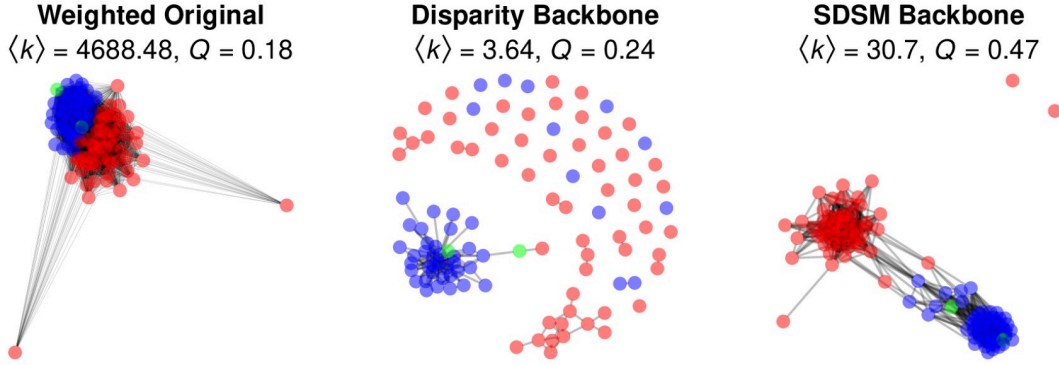

**Fig 5. Extracting the backbone of a bipartite projection of bill co-sponsorship in the 115th US Senate.**

with relying on a raw projection. First, the network has a high density and large weighted mean degree $\langle k \rangle$ = 4688.48 that obscures any structure. Second, the visualization fails to capture the known partisan polarization of the US Senate [5, 30, 32], which is confirmed by the network's small modularity ($Q$ = 0.18) with respect to political party affiliation. Finally, the network includes as connected two nodes representing Senators Jeff Sessions (R-AL) and Jon Kyl (R-AZ). Although these senators did co-sponsor some bills with their Republican colleagues, they co-sponsored very few because their terms were unusually short: Sessions served only 36 days before being appointed US Attorney General, while Kyl served only 117 days to fill a vacancy left by the death of Senator John McCain.

Because this bipartite projection is a weighted network, we could extract its backbone using the disparity filter:

```
> disparity <- disparity(P, alpha = 0.1, class = "igraph")
 This object looks like it could be a bipartite projection. If
so, consider extracting the backbone using a model designed for
bipartite projections: sdsm, fdsm, fixedfill, fixedrow, or
fixedcol.
```

The disparity backbone is shown in the middle panel of Fig 5. Even using a relatively liberal significance level ($\alpha$ = 0.1), this backbone is so sparse ($\langle k \rangle$ = 3.64) that the majority of Senators are disconnected. Additionally, although there are small clusters of Democrats and Republicans, its modest modularity ($Q$ = 0.24) highlights that it fails to capture the known partisan polarization. These problems arise because the disparity filter is applied directly to the bipartite projection, and is unable to incorporate the more detailed information contained in the original bipartite network.

Finally, we can extract a backbone using the stochastic degree sequence model:

```
> backbone <- sdsm(senate, alpha = 0.05, class = "igraph",
narrative = TRUE)
 === Suggested manuscript text and citations ===
 We used the backbone package for R (v2.1.0; Neal, 2022) to
extract the unweighted backbone of the weighted projection of an
unweighted bipartite network containing 105 agents and 3665
artifacts. An edge was retained in the backbone if its weight
was statistically significant (alpha = 0.05) using the stochas-
tic degree sequence model (SDSM; Neal, 2014). This reduced the
number of edges by 69.7%, and reduced the number of connected
nodes by 1.9%.
 Neal, Z. P. (2022). backbone: An R Package to Extract Network
Backbones. arXiv:
 2203.11055 [cs.SI]. https://doi.org/10.48550/
arXiv.2203.11055
 Neal, Z. P. (2014). The backbone of bipartite projections:
Inferring relationships from co-authorship, co-sponsorship, co-
attendance and other co-behaviors. Social Networks, 39, 84-97.
https://doi.org/10.1016/j.socnet.2014.06.001
```

By including the `narrative = TRUE` argument, the function generates text and citations describing the backbone extraction. The SDSM backbone is shown in the right panel of Fig 5. It is sparse ($\langle k \rangle$ = 30.7), but not as sparse as the disparity backbone. Accordingly, with the desirable exceptions of Sessions and Kyl, all the Senators are connected in a single component. Moreover, this connected component clearly illustrates the known partisan polarization of the US Senate, which is confirmed by the network's high modularity ($Q$ = 0.47).

## Backbones of unweighted networks

### Background

Backbone extraction from weighted networks and weighted bipartite projections is facilitated by the availability of information about edge weights. A different approach is required for extracting the backbone from unweighted networks because there are no edge weights. Many backbone models for unweighted networks have been proposed [33–38], but they all follow a common process: score, normalize, filter, connect:

1. Because there are no weights associated with the edges, each edge is assigned a *score*. These scores can be random [38], but more often are topologically-derived. For example, an edge may be assigned a score that counts the number of triangles [36] or quadrangles [33] it completes, or that captures the amount of overlap in the neighborhoods of the two nodes it connects [35, 37].

2. These edge scores can then be *normalized*. The normalization step is omitted by some backbone models [35, 38], while other backbone models normalize through rankings [34, 37] or more sophisticated transformations [33, 36].

3. The (optionally normalized) edge scored are *filtered* according to a user-specified sparsification parameter. Different backbone models interpret this sparsification parameter in different ways. For example, this parameter may specify the fraction of strongest scoring edges to retain [38], the threshold edge score above which edges are retained [33, 35, 36], or a non-linear parameter controlling the stringency of edge retention [34, 37].

4. Because unweighted backbone models can frequently yield disconnected networks, some models ensure the *connectedness* of the backbone by also including edges that appear in the union of minimum spanning trees [33].

Each of these steps in the process represent a decision point in the specification of a backbone model. The `sparsify()` function in the `backbone` package allows users to explicitly specify how (or whether) each step is performed, and therefore to customize a unique backbone model. However, some combinations represent named backbone models that have been investigated in the literature. In this paper, we focus on two such models that are fast and often yield informative backbones: *Local Sparsification* (L-Spar) [37] and *Local Degree* (LD) [34].

The L-Spar model [37] scores edges using the Jaccard coefficient, which measures the amount of overlap in the neighbors of the two nodes it connects, ranging from 0 (no overlap) to 1 (complete overlap). Here, the intuition is that relationships are stronger between individuals who share many of the same contacts, which makes this model most appropriate for extracting backbones that preserve clustering structures. Next, the edge scores are normalized by ranking them from the perspective of each node, such that a node's top scoring edge is ranked highest. Finally, for a node with degree $d$, the $d^s$ highest-ranked edges are preserved in the backbone, where $s$ is the sparsification parameter ranging from 0 to 1. When $s = 0$, which is the smallest value yielding the sparsest backbone, every node's one strongest edge is retained because $d^0 = 1$ for all $d$. Alternatively, when $s = 1$, which is the largest value yielding the densest backbone, all of a node's edges are retained because $d^1 = d$ for all $d$. Intermediate values of $s$ retain more or fewer edges, but non-linearly with the goal of removing edges of "nodes with higher degree more aggressively than nodes with lower degree" [37].

The LD model [34] scores edges, from the perspective of each node, using the degree of the node at the other end. Here, the intuition is that the most important edges are those that lead to hubs, which makes this model most appropriate for extracting backbones that preserve

branching, hub-and-spoke, or hierarchical structures. After this scoring step, the LD model proceeds identically to the L-Spar model: edges are normalized by rank, then the $d^s$ highest-ranking edges for each node are preserved in the backbone.

## Toy examples

To illustrate extracting backbones from unweighted networks using the `backbone` package, we begin with two separate toy examples:

**L-Spar and communities.** The L-Spar backbone model is designed to extract backbones that preserve hidden community structures. To illustrate, we begin by generating a random unweighted network embedded with a hidden community structure:

```
> pref.matrix <- matrix(c
(.75,.25,.25,.25,.75,.25,.25,.25,.75),3,3)
> unweighted <- sbm.game(60, pref.matrix, c(20,20,20))
```

The `sbm.game()` function is from the `igraph` [39] package, and is used to generate stochastic block models. In this case, it yields a 60-node unweighted, undirected network composed of three 20-node communities, such that there is a 75% chance of a within-community edge and a 25% change of a between-community edge. The left panel of Fig 6 shows the resulting network, which is so dense that the community structure is obscured.

We can extract the L-Spar backbone using either the flexible `sparsify()` function or the simpler `sparsify.with.lspar()` function:

```
> backbone <- sparsify(unweighted, escore = "jaccard", nor-
malize = "rank",
          filter = "degree", umst = FALSE, s = 0.5)
> backbone <- sparsify.with.lspar(unweighted, s = 0.5)
```

The `sparsify()` function is highly customizable by allowing the user to specify how to perform each of the four steps in an unweighted backbone model, while the `sparsify.with.lspar()` function is a simplified wrapper that performs these steps according to the L-Spar model. In either case, the sparsification parameter `s` specifies the stringency of the backbone model in preserving edges. The right panel of Fig 6 shows the L-Spar backbone, which clearly reveals the known three-community structure of this network.

# Unweighted Original     L–Spar Backbone

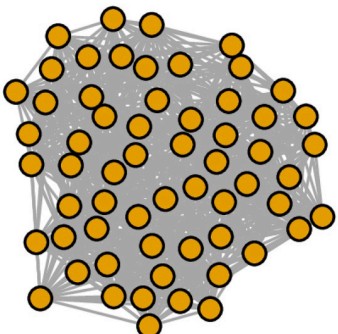 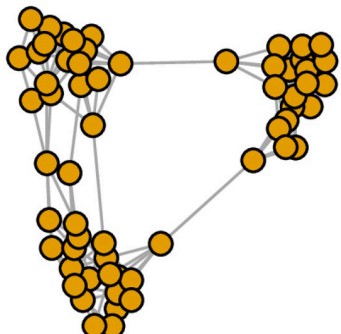

**Fig 6. Extracting the backbone of an unweighted network with embedded communities (toy example).**

**LD and hubs.** The LD backbone model is designed to extract backbones that preserve hidden hub-and-spoke or hierarchical structures. To illustrate, we begin by generating a random unweighted networkr embedded with hidden hubs:

```
> unweighted <- as.undirected(sample_pa(60, m = 3), mode =
"collapse")
```

The `sample_pa()` function is from the `igraph` [39] package, and is used to generate random networks formed via preferential attachment. In this case, it yields a 60-node unweighted, undirected network in which a few nodes occupy high-degree hub positions. The left panel of Fig 7 shows the resulting network, which is so dense that the hubs are obscured.

We can extract the L-Spar backbone using either the flexible `sparsify()` function or the simpler `sparsify.with.localdegree()` function:

```
> backbone <- sparsify(unweighted, escore = "degree", normal-
ize = "rank",
          filter = "degree", umst = FALSE, s = 0.1)
> backbone <- sparsify.with.localdegree(unweighted, s = 0.1)
```

Here, the `sparsify.with.localdegree()` function is a wrapper for the more flexible `sparsify()` function that performs each of the steps according to the local degree model. The right panel of Fig 7 shows the LD backbone, which clearly reveals the network's hierarchical structure around a few high-degree hubs.

### Empirical examples

To illustrate the extraction of backbones from unweighted networks in practice, we use two separate empirical examples:

**L-Spar and friendship.** To illustrate the use the L-Spar model to reveal a hidden community structure, we use data on the undirected and unweighted friendships among 79 faculty at a UK university [6]. Each individual is a member of a single school, which provides an expectation for a community structure. The left panel of Fig 8 shows the original network, with nodes colored according to their school membership. For a small network, it is relatively dense, with a mean degree $\langle k \rangle$ = 14.25, which can hamper the identification of communities. In this case, the communities are relatively obvious only because the nodes are already colored according to their known community memberships. However, communities detected by applying a fast-

# Unweighted Original     LD Backbone

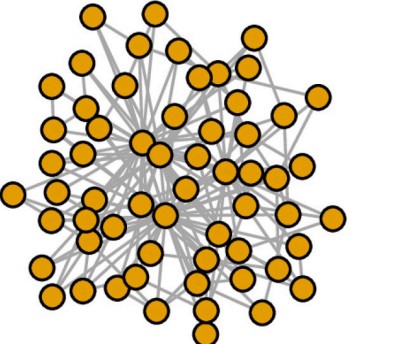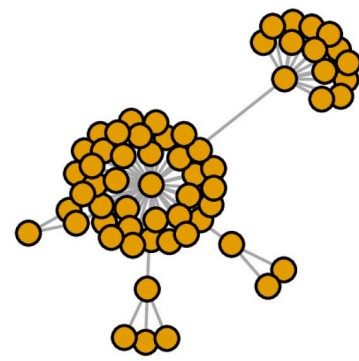

**Fig 7. Extracting the backbone of an unweighted network with embedded hub/spoke structure (toy example).**

## Unweighted Original
$\langle k \rangle = 13.97$, $ARI_{fg} = 0.76$

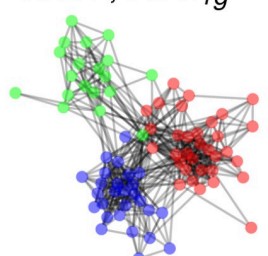

## L–Spar Backbone
$\langle k \rangle = 5.32$, $ARI_{fg} = 0.92$

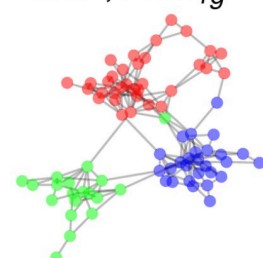

**Fig 8. Extracting the backbone of an unweighted network of faculty email exchanges.**

greedy algorithm to this network yields only a moderate recovery of the true memberships as measured by the Adjusted Rand Index ($ARI_{fg} = 0.76$) [40].

We can extract the network's L-Spar backbone using:

```
> backbone <- sparsify.with.lspar(faculty, s = 0.5, class =
"igraph", narrative = TRUE)
=== Suggested manuscript text and citations ===
We used the backbone package for R (v2.1.0; Neal, 2022) to
extract the unweighted backbone of an unweighted and undirected
unipartite network containing 79 nodes. Specifically, we used
Satuluri et al.'s (2011) L-Spar model with a sparsification
threshold of 0.5.
This reduced the number of edges by 62%, and reduced the num-
ber of connected nodes by 0%.
Neal, Z. P. (2022). backbone: An R Package to Extract Network
Backbones. arXiv:
2203.11055 [cs.SI]. https://doi.org/10.48550/
arXiv.2203.11055
Satuluri, V., Parthasarathy, S., & Ruan, Y. (2011, June).
Local graph sparsification for scalable clustering. In Proceed-
ings of the 2011 ACM SIGMOD International Conference on Manage-
ment of data (pp. 721-732). https://doi.org/10.1145/
1989323.1989399
```

By including the `narrative = TRUE` argument, the function generates text and citations describing the backbone extraction. The resulting backbone is shown in the right panel of Fig 8. It is substantially less dense, with a much smaller mean degree $\langle k \rangle = 5.32$, which more clearly reveals the three-community structure. Indeed, detecting communities by applying a fast-greedy algorithm to the backbone yields an excellent recovery of the true memberships ($ARI_{fg} = 0.92$) [40].

**LD and air traffic.** To illustrate the use the LD model to reveal hidden hubs, we return to the US air traffic data discussed earlier. From the original weighted network, we focus on the unweighted version:

```
> unweighted <- simplify(graph_from_adjacency_matrix(airport,
mode = "undirected",
                         diag = FALSE))
```

```
> unweighted["JFK","LAX"] #Are there flights between New York
and Los Angeles?
 [1] 1
> unweighted["LAN","LAX"] #Are there flights between Lansing
and Los Angeles
 [1] 0
> sum(unweighted["LAN",]) #How many destinations are reach-
able from Lansing?
 [1] 40
```

The `simplify()` function from the `igraph` [39] package transforms the originally weighted airline network into an unweighted network in which two airports are connected if *any* number of passagers flew between them. Examining this unweighted network, we observe that there are flights between JFK (New York) and LAX (Los Angeles), but not between LAN (Lansing) and LAX (Los Angeles), and indeed that it is possible to reach only 40 airports from LAN. The left panel of Fig 9 shows this unweighted network, which is so dense and has such a high mean degree ($\langle k \rangle$ = 50.67) that the underlying structure of the US airline infrastructure is obscured. Additionally, its degree scaling exponent $\gamma$ = 6.84 is inconsistent with the transportation network's expected in a hub-and-spoke transportation network [22, 23].

We can extract the network's LD backbone using:

```
> backbone <- sparsify.with.localdegree(unweighted, s = 0.3,
narrative = TRUE)
=== Suggested manuscript text and citations ===
 We used the backbone package for R (v2.1.0; Neal, 2022) to
extract the unweighted backbone of an unweighted and undirected
unipartite network containing 382 nodes. Specifically, we used
Hamann et al.'s (2016) local degree model with a sparsification
threshold of 0.3. This reduced the number of edges by 90.7%, and
reduced the number of connected nodes by 0%.
 Neal, Z. P. (2022). backbone: An R Package to Extract Network
Backbones. arXiv: 2203.11055 [cs.SI]. https://doi.org/10.48550/
arXiv.2203.11055
 Hamann, M., Lindner, G., Meyerhenke, H., Staudt, C. L., & Wag-
ner, D. (2016). Structure-preserving sparsification methods for
social networks. Social Network Analysis and Mining, 6(1), 22.
https::://10.1007/s13278-016-0332-2
```

## Unweighted Original
$\langle k \rangle$ = 50.67, $P(k)$ ~ $k^{-6.84}$

## LD Backbone
$\langle k \rangle$ = 4.71, $P(k)$ ~ $k^{-1.94}$

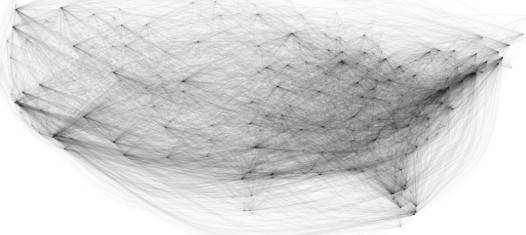 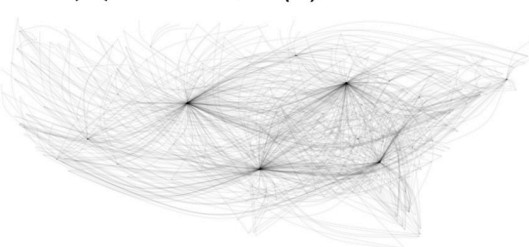

**Fig 9. Extracting the backbone of an unweighted network of continental US airline traffic in 2019.**

By including the `narrative = TRUE` argument, the function generates text and citations describing the backbone extraction. The resulting backbone is shown in the right panel of Fig 9. It is much sparser, with a smaller mean degree $\langle k \rangle$ = 4.71, which makes it easier to see the hub-and-spoke structure of the network anchored by the hub airports of ORD (Chicago), DEN (Denver), DFW (Dallas-Fort Worth), and ATL (Atlanta). Additionally, its degree scaling exponent $\gamma$ = 1.94 is consistent with a transportation infrastructure known to have a hub-and-spoke organization [22, 23].

## Issues in statistical inference

Some of the backbone models implemented in the `backbone` package are 'structural' models that choose edges to retain based on their topological properties. For example, the global threshold model makes such decisions on the basis of edge weights, while the L-Spar model makes such decisions on the basis of an edge's Jaccard coefficient, which functions as an imputed edge weight. However, other models are 'statistical' in the sense that they choose edges to retain by considering their probability relative to a statistical null model. These statistical models compute a *p*-value for each edge, where the *p*-value captures the probability of observing a larger weight associated with the same edge in a random network. The statistical models are identifiable from the presence of an `alpha` parameter; an edge is retained if its *p*-value is smaller than the specified $\alpha$-value. This section briefly reviews several issues related to statistical backbone models' *p*-values.

### Exact versus approximate *p*-values

Most statistical models can compute edges' *p*-values exactly using analytic methods [28]. However, the fixed degree sequence model used for extracting the backbone from bipartite projections and implemented in the `fdsm()` requires estimating these *p*-values using Monte Carlo simulation. This function uses efficient methods for simulating random networks under the FDSM null model [41], however it is still necessary to perform a large number of simulations to estimate *p*-values with sufficient confidence that they can be compared to the $\alpha$-value. The `trials` parameter of the `fdsm()` function allows the user to explicitly specify how many simulations are used to estimate the *p*-value. When the `trials` parameter is not specified, the function automatically determines the required number of simulations to estimate *p*-value with sufficient confidence. Using the US Senate data, for example:

```
> backbone <- fdsm(senate, alpha = 0.05, trials = 1000)
Constructing empirical edgewise p-values using 1000 trials -
|====================================================| 100%
> backbone <- fdsm(senate, alpha = 0.05)
Constructing empirical edgewise p-values using 80297 trials -
|====================================================| 100%
```

When `trials = 1000` is specified, the *p*-values are estimated from 1000 random bipartite networks, which takes only a few seconds but yields potentially unstable results. In contrast, when the `trials` parameter is omitted, `fdsm()` automatically determines that testing the *p*-values against a significance level of $\alpha$ = 0.05 will require 80,297 random bipartite networks, which takes several minutes. This highlights why the FDSM backbone model is often impractical for extracting the backbone from large bipartite projections, or at conservative significance levels (e.g., $\alpha < 0.05$).

## Obtaining *p*-values

Typically statistical backbone models evaluate edges' p-values and return an unweighted back-bone that only contains the edges deemed statistically significant. However, specifying `alpha = NULL` instead returns the *p*-values themselves, rather than the backbone they imply. For example, in the case of the US airport network:

```
> bb.object <- disparity(airport, alpha = NULL)
  This matrix object is being treated as a weighted undirected
network containing 382 nodes.
> bb.object$G [1:3, 1:3]
      ALB    ATL AVP
ALB    0 24349   13
ATL 24349    0 5761
AVP   13 5761    0
> bb.object$Pupper[1:3, 1:3]
                ALB          ATL          AVP
ALB 1.00000000000 0.00006619769 0.986723244
ATL 0.00006619769 1.00000000000 0.001924381
AVP 0.98672324370 0.00192438055 1.000000000
> bb.object$Plower[1:3, 1:3]
          ALB        ATL          AVP
ALB 1.00000000 0.9999338 0.01327676
ATL 0.99993380 1.0000000 0.99807562
AVP 0.01327676 0.9980756 1.00000000
> bb.object$model
[1] "disparity"
```

By specifying `alpha = NULL`, the `disparity()` function returns a backbone object that contains four items. First, it contains the original weighted network; here we see that 24,349 passengers flew between Atlanta (ATL) and Albuquerque (ALB). Second, it contains the upper-tail *p*-values, which capture the probability of observing an edge weight *as large or larger* in a random network. For example, we see that observing so many passengers flying between ATL and ALB is very unlikely in a random network, which offers evidence that the ATL-ALB edge is significant. Third, it contains the lower-tail *p*-values, which capture the probability of observing an edge weight *as small or smaller* in a random network. For example, we see that observing so *few* passengers flying between ALB and Scranton (AVP) is very unlikely in a random network. Finally, it contains a string indicating the backbone model that generated these *p*-values.

Returning a backbone object containing *p*-values can be useful in cases where computing the *p*-values is slow (e.g., using `fdsm()`), and backbones extracted using multiple $\alpha$ significance levels are desired. In such cases, backbone extraction is a two-step process: first generate a backbone object containing the *p*-values, then use `backbone.extract` to extract a backbone at the desired $\alpha$ significance level. For example,

```
> bb.object <- disparity(airport, alpha = NULL) #Compute p-
values
  > bb1 <- backbone.extract(bb.object, alpha = 0.001) #Backbone
at alpha = 0.001
  > bb2 <- backbone.extract(bb.object, alpha = 0.005) #Backbone
at alpha = 0.005
```

The first line computes the *p*-values for each edge in the airport network using the disparity filter model. Then, the second line extracts the backbone using a significance level of $\alpha$ = 0.001. The third line extracts a different backbone using the more liberal significance level of $\alpha$ = 0.005, which can be performed without needing to re-compute the *p*-values.

## Correcting *p*-values

The chosen $\alpha$ significance level defines the Type-I error rate, which in the backbone extraction context is the probability of deciding an edge has a larger weight than would be expected in a random network, when in fact it does not (i.e., a false positive). In practice, backbone extraction requires evaluating every edge with a non-zero weight, which inflates the Type-I error rate. For example, if $\alpha$ = 0.05, then the probability of incorrectly deciding a single edge is significant is 0.05. However, the probability of making at least one such incorrect decision across *m* edges is $1 - (1 - 0.05)^m$, which can dramatically inflate the risk of false positives even when extracting relatively small backbones. Performing a Multiple Test Correction (MTC), which involves adjusting the computed *p*-values, is necessary to control for this error inflation. The statistical backbone models implemented in the `backbone` package can perform any of the MTC corrections that are provided by R's `p.adjust()` function via the `mtc` parameter.

We can use the empirical airport data to examine the effect of correcting for multiple tests:

```
> backbone_nomtc <- disparity(airport, alpha = 0.001, class =
"igraph", mtc = "none")
> backbone_mtc <- disparity(airport, alpha = 0.001, class =
"igraph", mtc = "holm")
```

The left panel of Fig 10 shows the disparity filter backbone extracted from the weighted network using an edgewise error rate of $\alpha$ = 0.001. That is, each edge's *p*-value is compared to 0.001 and retained if it is smaller. This yields a backbone in which there is only a 0.001% chance that *any given retained edge* is, in fact, not significant. However, because 9678 edges' significance is tested, there is a $1 - (1 - 0.001)^{9678} \approx 99.99\%$ chance that at least one retained edge is not significant. The right panel of Fig 10 shows the disparity filter backbone extracted using a *familywise* error rate of $\alpha$ = 0.001, achieved by applying a Holm-Bonferroni correction to the *p*-values [42]. This yields a much sparser backbone in which there is only a 0.001% chance that *one or more retained edge* is, in fact, not significant. Although the MTC-corrected backbone is much sparser, it still displays the expected hub-and-spoke structure.

**Disparity without MTC**
$\langle k \rangle$ = 4.34, *P(k)* ~ $k^{-2.24}$

**Disparity with MTC**
$\langle k \rangle$ = 1.26, *P(k)* ~ $k^{-2.53}$

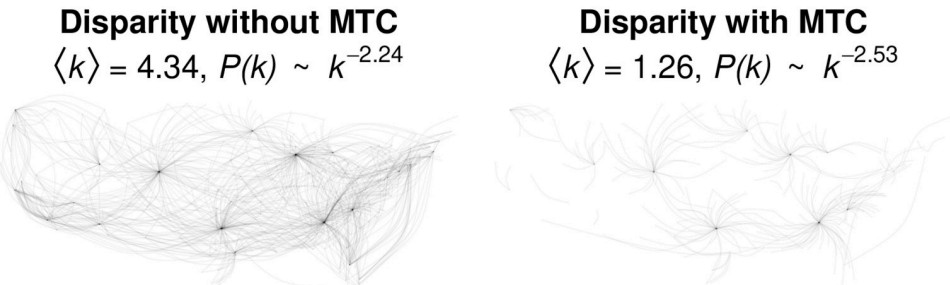

**Fig 10. Disparity filter backbone of a weighted network of continental US airline traffic in 2019 at a edgewise error rate (left) or familywise error rate (right) of $\alpha$ = 0.001.**

## Signed backbones

In most cases, backbone extraction involves identifying and retaining the statistically significantly *strong* edges in the backbone. However, statistical backbone models are also capable of identifying and retaining the statistically significantly *weak* edges. Specifying `signed = TRUE` yields a *signed* backbone in which significantly strong edges are retained in the backbone as positive edges, while significantly weak edges are retained in the backbone as negative edges. For example, returning to the US Senate data:

```
> signed <- sdsm(senate, alpha = 0.05, signed = TRUE)
```

Fig 11 shows the signed backbone, with positive edges drawn in green and negative edges drawn in red. This example illustrates that, as expected, positive relations of cooperation exist primarily between legislators from the same party, while negative relations of opposition exist primarily between legislators from different parties [5].

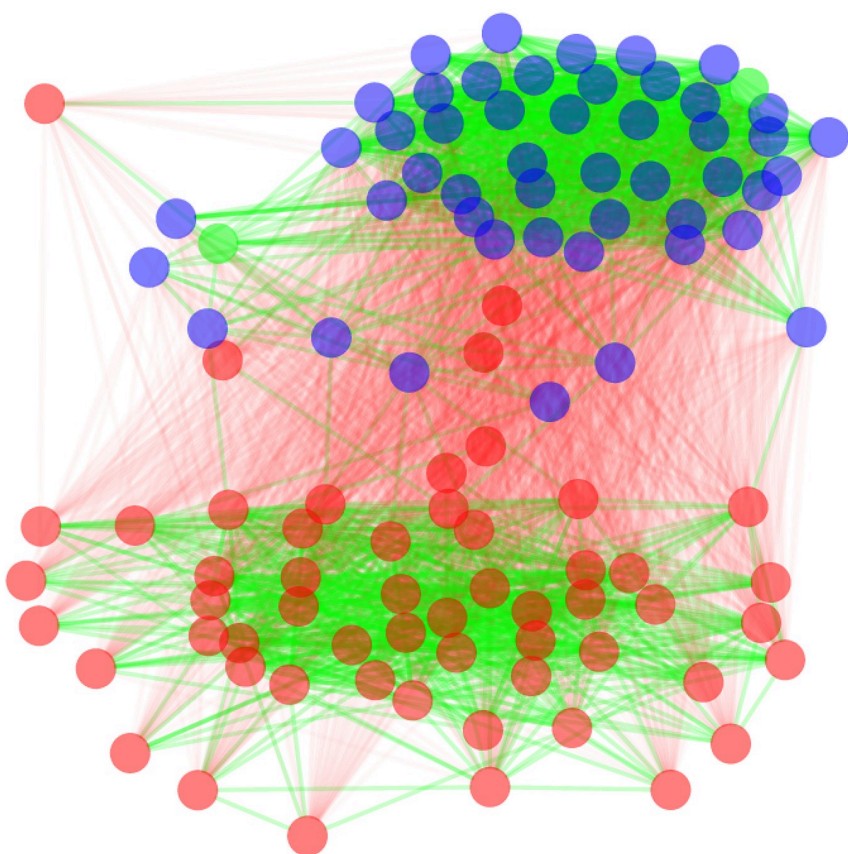

**Fig 11. Extracting the signed backbone of a bipartite projection of bill co-sponsorship in the 115th US Senate.**

Extracting an unweighted (i.e., not signed) backbone involves a one-tailed significance test, that is, testing whether an edge's observed weight is larger than the weight of the corresponding edge in random networks generated under the null model. In a one-tailed significance test, and edge is deemed significant if $p < \alpha$. However, extracting a signed backbone involves a *two*-tailed significance test because it requires evaluating whether an edge's observe weight it larger *or* smaller than expected in a random network. In such two-tailed significance tests, an edge is deemed significant and negative if its lower-tail $p < \frac{\alpha}{2}$, and is deemed significant and positive if its upper-tail $p < \frac{\alpha}{2}$.

## Conclusion

### Applications

The models implemented in the `backbone` package are generic and can be used to extract the backbone of networks observed in many different domains. To date, the majority of applications have been in political, developmental, and biological sciences. In political science, the `backbone` package has been used to infer networks of political alliances among legislators from a bipartite projection of their bill sponsorships [30–32, 43–45]. While two legislators may be viewed as having an alliance when they are observed to sponsor many of the same bills (i.e., the edge weight in a co-sponsorship network), backbone models offer a way to identify pairs of legislators that have sponsored significantly more bills together than expected. In developmental science, the `backbone` package has been used to measure childrens' social networks from their peer-reported or observed group memberships [46, 47]. This is useful because data on childrens' social relations are difficult to collect directly, but they can often be inferred from bipartite projections. In the biological sciences, the `backbone` package has been used to examine organism co-occurrences in both plants [48] and animals [49], and is used by the `genetonic` package [50] to extract the backbone of gene-geneset graphs. Beyond these scientific applications, the `backbone` package has also been used to examine the collaboration relations among the writers, pencilers, and editors responsible for the X-men comic book series [51], and to recommend holiday movies [52]. These applications highlight the flexibility of the backbone extraction functions implemented in the `backbone` package and offer additional illustrations of its functionality.

### Limitations and future work

The `backbone` package remains under development, and the models it implements are relatively new, which means that both the package and models are subject to some limitations that highlight opportunities for future work. First, many backbone models exist for both weighted and unweighted [7–17, 53–55] networks that are not yet implemented in the `backbone` package. However, the `backbone` package provides a generic framework for the implementation of additional models in future releases. Second, relatively little research has focused on validating backbone models or establishing their scope conditions, which leaves researchers with limited guidance on model selection. The `backbone.suggest()` function already provides rudimentary model selection assistance. However, by implementing multiple models in a single framework, the `backbone` package can facilitate future work comparing backbone models on data with a ground truth (for validation) or with different characteristics (to establish scope conditions). Finally, because the goal of backbone extraction is simplification, backbone models are most useful in very large networks, however certain backbone models computationally intensive. Thus, future work on backbone models and their implementations should focus on scalability and efficiency.

## Author Contributions

**Conceptualization:** Zachary P. Neal.

**Data curation:** Zachary P. Neal.

**Formal analysis:** Zachary P. Neal.

**Funding acquisition:** Zachary P. Neal.

**Investigation:** Zachary P. Neal.

**Methodology:** Zachary P. Neal.

**Project administration:** Zachary P. Neal.

**Resources:** Zachary P. Neal.

**Software:** Zachary P. Neal.

**Validation:** Zachary P. Neal.

**Visualization:** Zachary P. Neal.

**Writing – original draft:** Zachary P. Neal.

**Writing – review & editing:** Zachary P. Neal.

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
