## [Decision Letter · Decision Letter 0]

1 May 2022

PONE-D-22-07882backbone: An R package to extract network backbonesPLOS ONE

Dear Dr. Neal,

Thank you for submitting your manuscript to PLOS ONE. After careful consideration, we feel that it has merit but does not fully meet PLOS ONE’s publication criteria as it currently stands. Therefore, we invite you to submit a revised version of the manuscript that addresses the points raised during the review process.

We look forward to receiving your revised manuscript.

Kind regards,

Hocine Cherifi

Academic Editor

PLOS ONE

Journal Requirements:

"This work was supported by the National Science Foundation (#1851625 & #2016320). The author thanks Rachel Domagalski and Bruce Sagan for their contributions to earlier versions of the backbone package."

We note that you have provided funding information. However, funding information should not appear in the Acknowledgments section or other areas of your manuscript. We will only publish funding information present in the Funding Statement section of the online submission form. 

"ZN received funding from the National Science Foundation (www.nsf.gov), awards #1851625 and #2016320. The funders had no role in study design, data collection and analysis, decision to publish, or preparation of the manuscript."

Reviewers' comments:

Reviewer's Responses to Questions

**Comments to the Author**

1. Is the manuscript technically sound, and do the data support the conclusions?

Reviewer #1: Yes

Reviewer #2: Yes

Reviewer #3: Yes

2. Has the statistical analysis been performed appropriately and rigorously? 

Reviewer #1: Yes

Reviewer #2: Yes

Reviewer #3: Yes

3. Have the authors made all data underlying the findings in their manuscript fully available?

Reviewer #1: Yes

Reviewer #2: Yes

Reviewer #3: Yes

4. Is the manuscript presented in an intelligible fashion and written in standard English?

Reviewer #1: Yes

Reviewer #2: Yes

Reviewer #3: Yes

5. Review Comments to the Author

Reviewer #1: The paper is a manual for the use of the backbone R package: a piece of software to extract salient edges from complex networks.

I love many things about this paper/package, among them being a built-in way to do multiple test corrections, or the possibility of returning signed backbones.

There are a few things that could make the paper (and the package) better.

The limitation section is honest about the lack of implementation for many methods used in the literature. I feel the author really should add the following to the list:

- Slater. A two-stage algorithm for extracting the multiscale backbone of complex weighted networks. PNAS, 2009

- Grady et al. Robust classification of salient links in complex networks. Nature communications, 2012

- Gemmetto et al. Irreducible network backbones: unbiased graph filtering via maximum entropy. arXiv, 2017

- Coscia & Neffke. Network backboning with noisy data. ICDE, 2017

- Šubelj. Convex skeletons of complex networks. Journal of The Royal Society Interface, 2018

Adding alternatives is important because, e.g., the disparity filter uses a very specific null model that might not apply to one's data -- e.g. it tends to create hub-and-spoke structures because hubs tend to connect more strongly, like Fig 3 illustrates, at the penalty of more decentralized community structures, which are also a feature of many real world networks. If these methods are cited, it reassures the reader that they are in the radar and will be added soon. Many of the papers I mention have dozens of citations and they are widely used, a few works even for unweighted networks.

As of now, I can say that this paper is good, and the backbone package is good, but I'm not using it, because most of the times I use backboning methods that are not implemented in it. If they are not part of the planned development, I don't think I will make the investment of learning to use the package.

In the introduction the author says that the backbone is an unweighted network, but the paper also mentions that it can consider "p-values as edge weights". However, some methods can re-normalize weights (like, e.g., Slate or Grady) or use z- or t-scores. I'd use a different language in the introduction, because as of now it seems the package returns exclusively unweighted backbones, which appears to be a severe limitation.

In the background section the author fails to mention that often, especially in graph theory, edge weights might represent costs, and not capacities. If a network has such weights, a backbone method that assumes edge weights being capacities would select the wrong edges for the backbone. This problem should be mentioned to make sure the user understands the assumption baked up in the package -- and perhaps having a function "costs2capacities" in the package could be useful.

An important information the package should also return is the fraction of nodes dropped from the backbone because they became isolates -- they have no edges in the backbone. Global threshold and disparity filter tend to isolate lots of nodes, while some of the cited methods don't.

Minor:

- Some typos/grammar mistakes (e.g. "an edge is retained if it's p-value is smaller than" -> "its p-value")

Reviewer #2: This contribution describes some usage scenarios for the ‘backbone’ package of the R suite to obtain the backbone representation of input networks. The use is demonstrated for methods for the extraction of the backbone from weighted networks, weighted bipartite projections, and unweighted networks.

The presentation is very clear with few relevant examples taken from public sources and made available also through the author's github repository. The meaning and importance of selected relevant parameters are addressed in good detail.

Overall I think this will be a very useful publication especially in the context of training students/postdocs in the analysis of network as well as to enable less expert statisticians to make some fundamental analysis. I have the following suggestions:

1. the abstract mentions that "This paper introduces [...] the `backbone' package for R,". I am not sure what is meant here - the package has already been published elsewhere.

2. in light of their relevance throughout the paper, the author might want to explain in one paragraph what are logarithmic degree distribution plots and the associated parameters

3. to support adoption by teachers, it would be quite useful if the author provided other example datasets that are not demonstrated in the paper but could be analysed fruitfully using the 'backbone' package (e.g. in a hands-on lab session)

Reviewer #3: The authors introduce a backbone package which implements, in R, methods for extracting the backbone (the most relevant edges and nodes in a network ) from weighted networks, weighted bipartite projections, and unweighted networks. The backbone extraction package was applied on empirical networks from different fields such as politicians' networks, social networks, and biological networks. This paper shows also the flexibility and the functionality of the backbone extraction functions implemented in the backbone package.

This article is certainly relevant for the readership of 'PLOS one' and I am confident that, once the authors will have addressed the suggestions below, it will be a suitable article for publication.

The authors seem to be unaware of recent works on backbone extraction methods using information about the community structure of the network worth quoting:

- Ghalmane, Z., Cherifi, C., Cherifi, H., & El Hassouni, M. (2021). Extracting modular-based backbones in weighted networks. Information Sciences, 576, 454-474.

and :

- Ghalmane, Z., Cherifi, C., Cherifi, H., & El Hassouni, M. (2020). Extracting backbones in weighted modular complex networks. Scientific Reports, 10(1), 1-18.

6. PLOS authors have the option to publish the peer review history of their article (what does this mean?). If published, this will include your full peer review and any attached files.

Reviewer #1: No

Reviewer #2: **Yes: **Antonio Rosato, Ph.D.

Reviewer #3: **Yes: **Zakariya Ghalmane

---

## [Editor Report · Decision Letter 1]

16 May 2022

backbone: An R package to extract network backbones

PONE-D-22-07882R1

Dear Dr. Neal,

We’re pleased to inform you that your manuscript has been judged scientifically suitable for publication and will be formally accepted for publication once it meets all outstanding technical requirements.

Kind regards,

Hocine Cherifi

Academic Editor

PLOS ONE
---

## [Editor Report · Acceptance letter]

18 May 2022

PONE-D-22-07882R1 

backbone: An R package to extract network backbones 

Dear Dr. Neal:

I'm pleased to inform you that your manuscript has been deemed suitable for publication in PLOS ONE. Congratulations! Your manuscript is now with our production department. 

Kind regards, 

on behalf of

Professor Hocine Cherifi 

Academic Editor

PLOS ONE